# Label-Free Detection of DNA via Surface-Enhanced Raman Spectroscopy Using Au@Ag Nanoparticles

**DOI:** 10.3390/nano12183119

**Published:** 2022-09-08

**Authors:** Ting Zhang, Xubin Quan, Naisi Cao, Zhaoying Zhang, Yang Li

**Affiliations:** 1Department of Pharmaceutical Analysis and Analytical Chemistry, Research Center for Innovative Technology of Pharmaceutical Analysis, College of Pharmacy, Harbin Medical University, Harbin 150081, China; 2The Fourth Hospital of Harbin Medical University, Harbin 150001, China

**Keywords:** surface-enhanced Raman spectroscopy, Au@Ag nanoparticles, DNA, label-free detection

## Abstract

DNA is a building block of life; surface-enhanced Raman spectroscopy (SERS) has been broadly applied in the detection of biomolecules but there are challenges in obtaining high-quality DNA SERS signals under non-destructive conditions. Here, we developed a novel label-free approach for DNA detection based on SERS, in which the Au@AgNPs core–shell structure was selected as the enhancement substrate, which not only solved the problem of the weak enhancement effect of gold nanoparticles but also overcame the disadvantage of the inhomogeneous shapes of silver nanoparticles, thereby improving the sensitivity and reproducibility of the SERS signals of DNA molecules. The method obtained SERS signals for four DNA bases (A, C, G, and T) without destroying the structure, then further detected and qualified different specific structures of DNA molecules. These results promote the application of SERS technology in the field of biomolecular detection.

## 1. Introduction

DNA contains genetic information and is the main material basis of biological inheritance. The detection and analysis of DNA structures play crucial roles in life sciences, including disease prevention, diagnoses, treatments, and drug development [1,2,3,4,5]. Traditional DNA detection methods, such as high-performance liquid chromatography–mass spectrometry (HPLC–MS), polymerase chain reaction (PCR), fluorescent labeling, molecular hybridization, and other strategies have been used [6,7,8,9,10]. However, most of the above methods are limited by high costs, low sensitivity, and complicated operations, which tend to lose the epigenetic information of DNA and cannot effectively detect the complete information of the DNA [11,12,13,14,15]. Therefore, the development of novel DNA detection techniques is a challenge.

In the late 1960s, with the development of life science techniques, Raman spectroscopy was increasingly applied by researchers to explore and analyze biological molecules and life processes due to its advantages (e.g., fast detection, high sensitivity, and low sample consumption) [16,17,18]. Raman spectroscopy is able to provide specific fingerprinting of different biomolecules at the molecular or lower level, but it is difficult to obtain the signals of DNA molecules directly due to the extremely weak intrinsic Raman signals of nucleic acid molecules, which often require extremely high concentrations to obtain reliable signals of DNA molecules. The emergence of surface-enhanced Raman scattering (SERS) has provided new opportunities for detecting molecular structures of nucleic acids [19,20,21,22,23,24], mainly in the prepared surfaces of good metallic conductors, where the enhanced electromagnetic field (or near the surface of the sample results) enhance the Raman scattering signal of the adsorbed molecules compared to the normal Raman scattering signal [25,26,27,28]. In a previous publication by our group, structural data on DNA i-motifs and DNA G-quadruplexes were successfully detected and analyzed using SERS spectroscopy [29,30].

The rapid development of nanotechnology has driven the widespread application of SERS technology. Numerous studies have indicated that surface plasmon resonance (SPR) has induced local electromagnetic field enhancement; it is considered one of the major roles in the SERS mechanism, and the plasmonic properties of different types of metallic nanoparticles influence SERS tremendously [31,32,33,34,35,36]. Among these, silver nanoparticles (AgNPs) and gold nanoparticles (AuNPs) are two of the most commonly used substrates for SERS enhancement, mainly because of the SPR effect in the visible and near-infrared range [37]. They have their own advantages and disadvantages (as SERS substrates), such as AuNPs, which are easy to prepare and tunable in structure and size, but the SPR effect is weak. In contrast, AgNPs can produce stronger and more sensitive SERS signals, but their sizes are difficult to control in the preparation process. In addition, the SERS-enhanced effect is dependent on the structure, size, shape, elemental composition, and surrounding environment of the nanoparticles [38,39,40]. Therefore, various morphologies of nanoparticles have been designed as substrates for SERS enhancement, such as nanospheres, nanostars, nanosheets, nanoflowers, etc. [41,42,43,44,45,46,47]. Of these, spherical nanoparticles are relatively stable as compared to metallic nanoparticles; the Au@Ag core–shell structure can greatly enhance the SERS signal due to the bimetallic synergistic effect and achieve detection stability and reproducibility. Liu et al. designed a SERS assay based on Au@Ag core–shell nanoparticles for milk insecticide amitraz residues [48]. Gao’s group synthesized porous Au@Ag nanospheres with highly dense and accessible hot spots for the SERS analysis [49]. However, Au@Ag nanoparticles as reagents for DNA structure detection based on SERS have rarely been reported.

Herein, we present a novel method for the label-free detection of DNA via a SERS technique. As illustrated in Figure 1, Au@Ag core–shell nanoparticles (Au@AgNPs) were first synthesized and obtained by centrifugation. The background Raman signal of the substrate was removed by washing and incubating with I^−^. The Raman effect was enhanced by mixing Au@AgNPs with an aqueous DNA solution and adding Ca^2+^ as an aggregating agent to form ‘‘hot spots’’. We explored the quantitative analysis of different concentrations of the same double-stranded DNA (dsDNA) structure and single bases of different sequences of dsDNA. Moreover, we further researched the sensitive identification and differentiation of different structures of dsDNA, triple-stranded DNA (tsDNA), and G-quadruplex. Hence, the method of the unlabeled detection of DNA structures based on SERS has potential application in DNA sequencing, gene detection, and early diagnosis of diseases.

## 2. Materials and Methods

### 2.1. Materials

Chloroauric acid was purchased from Beijing West Asia Chemical Company, Ltd., Beijing, China. Silver nitrate and sodium citrate dihydrate (C_6_H_5_Na_3_O_7_∙2H_2_O) were obtained from Alfa Aesar (China) Chemical Company Limited, (Shanghai, China), and ascorbic acid was purchased from Tianjin Beilian Fine Chemicals Development Co., Tianjin, China. All solutions were prepared with deionized water. The glassware was washed with aqua regia (HCl:HNO_3_ = 3:1 (*v*/*v*)) and rinsed several times with pure water before the experiments.

### 2.2. Instruments

TEM images were examined by a Hitachi HT7700 all-digital 120 kV transmission electron microscope. HADDF-STEM was conducted using Thermo Scientific Talos F200S. DLS was recorded on a Malvern Instruments, Zetasizer APS. CD spectrum was measured by a BioLogic Science Instruments MOS-450 spectrometer.

### 2.3. Preparation of Au Seeds

Au seeds with an average diameter of approximately 20 nm were prepared using a reduction of HAuCl_4_ by sodium citrate dihydrate. Briefly, 0.060 g of sodium citrate dihydrate was added to 400 mL of purified water in a 500 mL three-necked flask, and the solution was heated to a slight boil under strong magnetic stirring, followed by the addition of 120 uL of HAuCl_4_ (250 g/L), which continued to boil and then stopped. The colorless solution gradually changed to burgundy, and the resulting solution was used as Au seeds for the synthesis of core–shell Au@AgNPs.

### 2.4. Preparation of Core–Shell Au@AgNPs

The core–shell Au@AgNPs were synthesized using a seed-mediated method. Briefly, the above synthesized AuNPs were sonicated for 10 min with an ultrasonic cleaner (KM-23C, Kemun Cleaning Technology Co., Ltd., Guangzhou, China) to prevent aggregation of AuNPs. A total of 18 mL of ascorbic acid (10 mM) was mixed with 150 mL of the above AuNPs for 6 min. Then, 7.5 mL of aqueous silver nitrate solution (10 mM) was added dropwise to the above solution under strong magnetic stirring at a rate of one drop per 15 s. The Au@AgNPs were synthesized when the solution gradually changed from burgundy to orange with continuous stirring for 30 min. All experiments were carried out at 25 °C.

### 2.5. SERS Detection DNA Based on Au@AgICNPs

The specific experimental procedure for the preparation of Au@AgICNPs (with calcium ions as aggregating agents to enhance the substrate) was as follows: 10 uL of 1 mM potassium iodide solution was added to 10 uL of centrifuged Au@AgNPs. After incubation at room temperature for 6 h, different DNA solutions were added to the Au@AgINPs and 0.6 uL of 0.01 M calcium chloride solution was added. The system was shaken uniformly and analyzed by Raman spectroscopy. The detection was carried out at an excitation wavelength of 532 nm and a sweep of 25 mV for 30 s.

## 3. Results and Discussion

### 3.1. Characterization of Au@AgNPs

The Au@AgNPs were synthesized according to the seed growth method [40]. The transmission electron microscopy (TEM) image showed that Au@AgNPs were homogeneously dispersed with a particle size of approximately 45 nm (Figure 1a). The potential value of Au@AgNPs measured by the Zeta potential was −20.1 mV (Appendix A). The dynamic light scattering (DLS) data demonstrated a radius of 45.37 nm for the nanoparticles (Appendix A), which is consistent with the TEM data. In addition, the distributions of Au and Ag atoms on the atomic scale were analyzed by high-angle annular dark field-scanning transmission electron microscopy (HAADF-STEM). As shown in Figure 1b. The core–shell structure was also evident from the HAADF-STEM images, due to the large difference in the atomic number between Au (Z = 79) and Ag (Z = 47). Moreover, the structures of Au@AgNPs were confirmed by EDS elemental mapping, as demonstrated in Figure 1c,d, where the purple and green colors represent the spatial distributions of Au and Ag elements, respectively, and the core–shell structure of the nanoparticles could be visualized clearly by merging the images, with a shell thickness of approximately 10 nm (Figure 1e). Furthermore, a broad absorption peak at 400 nm was observed for Au@AgNPs through UV–Vis absorption spectroscopy (Figure 1f). This was attributed to the silver shell encapsulated on the gold surface, which is consistent with previous literature [45]. Taken together, the above experimental data jointly demonstrate the successful preparation of Au@AgNPs with a core–shell structure.

### 3.2. Reproducibility and Quantification Detection of the Same Sequence dsDNA

To demonstrate the ability of our newly developed substrate to enhance the SERS signal of dsDNA, we prepared dsDNA (D3: 5′-AGCTAGCTAGCTAGCTAGCT-3′ 5′-AGCTAGCTAGCTAGCTAGCT-3′) and characterized it by circular dichroism (CD). As shown in Appendix A, the result of CD demonstrated that the dsDNA structure was successfully synthesized, in agreement with the published literature [50]. Figure 2a shows the SERS spectra of D3 under different conditions. From the experimental results, the D3 was unable to detect the Raman signal (red line), but the substrate Au@AgNPs had a weak Raman signal (green line). To avoid the Raman effect of the substrate itself, it was successfully eliminated by washing the Au@AgNPs and adding I^−^ incubation to form Au@AgINPs (brown line). When we used Au@AgINPs as the enhanced substrate to detect the D3 sequence, we found that the signal was weak (blue line). Therefore, Ca^2+^ was introduced as an aggregation agent to induce Au@AgINPs to form Au@AgICNPs, producing ‘‘hot spots’’ to significantly enhance the Raman signal (pink line). The detected D3 Raman signal was consistent with the base signature peaks of dsDNA reported in the literature [51,52,53,54] (724 cm^−1^ = dA ring br, 1323 cm^−1^ = dG vs.(C-N) δ(C8-H) 1361 cm^−1^ = dG C2′-endo/syn, 1646 cm^−1^ = dT, dC vs. C5 = C6). On this basis, we also filtered the aggregators (Appendix A); the results showed that when Ca^2+^, Mg^2+^, and Zr^4+^ were used as aggregators, respectively, Ca^2+^ had the best enhancement effect and less effect on the dsDNA structure. The above results proved that Au@AgICNPs could significantly enhance the Raman effect and achieve label-free detection of dsDNA.

Gold nanoparticles (AuNPs) are common substrates for SERS detection of DNA, so we chose AuNPs for the control experiment to investigate the enhancement effects of Au@AgICNPs for SERS. As depicted in Appendix A, pure AuNPs were synthesized at a particle size of ~50 nm, which was similar to that of Au@Ag. When the D3 was detected simultaneously, as shown in Appendix A, the SERS enhancement effect of Au@AgICNPs with D3 was more obvious (red line), likely due to the bimetallic synergistic effect of the Au@Ag core–shell structure that greatly enhanced the SERS signal. Specifically, the SPR effects of metal nanoparticles are highly dependent on the geometry of the plasma structure, elemental composition, and surrounding environment. In general, the preparation process of AuNPs is simple and easy to control, but the enhancement effect of AuNPs is much weaker than that of AgNPs. In contrast, AgNPs can produce stronger, more sensitive SPR effects, but their sizes and shapes are not homogeneous. The Au@AgNPs we developed overcame the weak AuNP enhancement factor, and solved the problem of uneven AgNP shapes, achieving a bimetallic enhancement effect. In addition, according to previous literature reports, the binding of dsDNA to AuNP is not as easy as the binding of AgNPs [55]. Therefore, Au@AgNPs also improve the adsorption of nanoparticles by dsDNA. In summary, we innovatively observed a method for the label-free detection of dsDNA; the enhancement effect of this approach is significantly superior to that of the conventional AuNPs.

To assay the stability and reproducibility, a random SERS spectrum of 15 sets of D3 sequences at different time intervals is given (Figure 2b). It was found from the spectra that the detection of the D3 sequences with this substrate was highly reproducible (RSD = 0.0078), likely due to the homogeneous and stable sizes of the prepared Au@AgNPs. Another reason is that due to Ca^2+^ aggregates being acidic, the hydrogen ions preferentially bind to the phosphate backbone of dsDNA, which stabilizes the folding of the dsDNA and allows the substrate to be stably wrapped by the phosphate group. As a result, a stable enhancement signal is achieved as it stays out of the state of being randomly laid on the surface of the enhancing substrate.

In addition, we studied the correlation between the change in the concentration of the D3 sequence and the intensity of the Raman peak, further validating the stability and repeatability of the approach. Figure 2c reveals the SERS spectra of D3 at different concentrations in an aqueous solution. We observed that the intensity of the characteristic peak gradually increased with increasing D3 concentrations. As shown in Figure 2d, we evaluated the impacts of changes in the D3 concentration on the peak intensity using the cytosine and thymine (C + T) peaks of D3 at 789 cm^−1^, which showed an incredibly linear relationship (the peak of acetonitrile at 923 cm^−1^ was used as the internal standard). A good linear relationship was observed (R^2^ = 0.9960), and the error bar thresholds were much smaller than that needed to differentiate various concentrations. The above data suggest that the novel SERS assay using Au@AgICNP as a substrate is feasible for the identification of dsDNA.

### 3.3. Quantitative Detection of Single Bases of dsDNA

Further quantification of single bases of different lengths of dsDNA was performed; we first synthesized dsDNA of different lengths: D1 (5′-AGCTAGCT-3′ 5′-AGCTAGCT-3′), D2 (5′-AGCTAGCTAGCT-3′ 5′-AGCTAGCTAGCT-3′), D3, and D4 (5′-AGCTAGCTAGCTAGCTAGCT-3′ 5′-AGCTAGCTAGCTAGCTAGCT-3′), and characterized the dsDNA by the CD spectrum (Appendix A), which was in agreement with the characteristic CD peaks previously reported on in the literature [46], demonstrating the successful synthesis of different lengths of dsDNA.

Figure 3a,c show the SERS spectra of D1, D2, D3, and D4, with the intensity of the characteristic peaks gradually increasing as the length of dsDNA increased. In particular, the intensity of the characteristic peak at 789 cm^−1^ for C + T increased with the number of C and T bases. Likewise, the intensity of the characteristic peak for adenine (A) at 733 cm^−1^ increased with the number of A bases. We quantified the peak intensities of C + T bases at 789 cm^−1^ (Figure 3b); the error analysis was performed on the characteristic peak intensities of C + T and a good linear relationship was observed (R^2^ = 0.9891). Moreover, we quantified the peak intensity of the A base at 733 cm^−1^ (Figure 3d), a good linear relationship was also obtained (R^2^ = 0.9837). This was due to the strong interaction of acidic Ca^2+^ with DNA, which allowed no exposure of phosphate groups, thus maintaining the structural stability of the DNA molecule. The above results for dsDNA show that the developed method is suitable for the quantification of dsDNA in an aqueous solution, including the quantification of the concentration of the same sequence and the number of bases in different sequences; it holds great promise for clinical applications.

### 3.4. Time Stability Detection of dsDNA Base on Au@AgINPs

To demonstrate the temporal stability of Au@AgINPs for detection, we mixed Au@AgINPs with D3 (dsDNA) at different times (0, 0.5, 1, 2, 4, and 6 h) and collected SERS signals to observe the changes in the SERS spectra of D3 (Figure 4a). We found that a stable SERS signal could still be detected after mixing the Au@AgINPs substrate with double-stranded DNA D3 for a period of time. Afterward, since the peak intensity at 789 cm^−1^ of the D3 strand was easily detectable and stable, the time stability of the substrate for the detection of dsDNA was further confirmed. Therefore, to further illustrate the temporal stability of the base pair for the detection of dsDNA, the peak intensity values at 789 cm^−1^ of the characteristic peak of the C + T base of the D3 strand in the SERS spectrum were selected for comparison. Notably, the peak intensities of the SERS spectra of the C + T bases of D3 did not change significantly as time was extended but remained relatively stable (Figure 4b). From Figure 4c, it can be observed that the SERS peak intensity value of the A base of the D3 chain at 733 cm^−1^ remained stable for 6 h, so it is even more indicative that this SERS substrate has better time stability; it showed that the SERS signal did not change its performance over time but instead maintained a high measurement level over a range of time.

### 3.5. Detection of Different DNA Structures

To explore the detection and differentiation of varying DNA structures, we designed tsDNA and G-quadruplex, which were characterized by CD spectrum (Appendix A) and proved to be successfully synthesized [56]. Figure 5 shows the SERS spectra obtained for dsDNA, tsDNA, and G-quadruplex; the results proved that the characteristic peaks of the three different structures of DNA were successfully detected, which coincides with previous literature. As shown in Appendix A, according to the previous literature [57,58], tsDNA: vs. OPO; dT ring br = 793 cm^−1^, vs. PO_2_^−^, bk = 1089 cm^−1^, dT vs. C5 = C6 = 1630 cm^−1^; G-quadruplex: vs. OPO; dT ring br = 800 cm^−1^, vs. PO_2_^−^, bk = 1090 cm^−1^, dT, dG δ NH(N2) = 1264 cm^−1^. For the analysis of the characteristic peak positions, we compared the characteristic peak positions of the synthesized tsDNA and the dsRNA and found that since the tsDNA we synthesized did not have vs. OPO, dG δ R5, dG C2′-endo/syn, the characteristic peak positions of tsDNA and dsDNA had large degrees of differentiation here. Similarly, we compared the characteristic peak positions of the synthesized G-quadruplex and dsDNA and found that since the G-quadruplex we synthesized had dG δ NH(N2) and dG vs. C = O (O6 interbase H-bond), the characteristic peak positions of G-quadruplex and dsDNA had large differences here. Thus, it is shown that the characteristic peaks of DNA detected by our newly developed method are almost identical to those reported in the literature. Moreover, the SERS spectra generated with Au@AgICNPs as the substrate can be distinguished to the naked eye, demonstrating the ultra-high sensitivity of the method for the detection of different DNA structures. In addition, we performed a principal component analysis of 25 sets of spectra of dsDNA D3, tsDNA, and G-quadruplex using the PCA assay. Ultimately, the PCA experiment produced multiple points on PC1 and PC2 (Figure 5d). The multiple projection points of the three different structures of DNA were surrounded by a well-defined ellipse, which in each case did not overlap. Remarkably, the three different structures of DNA can be easily distinguished in an aqueous solution based on the variations in the positions of the characteristic peaks. The results show that DNA with different structures can be detected and distinguished by our method and that further rapid identification of DNA with different structures can be achieved by the PCA assay.

## 4. Conclusions

In summary, a novel method for the label-free detection of DNA based on SERS was successfully developed. The problems of low yields and impurities were solved by the new method of synthesizing Au@AgNPs with a core–shell structure. When Au@AgNPs were used as SERS-enhanced substrates, which enabled the determination of four bases in DNA, the base content of the DNA molecule was quantified. Moreover, in this paper, different specific structures of DNA were detected and qualitatively analyze. The method achieves high sensitivity and reproducibility of the SERS signal for DNA molecules without destroying the DNA structure. The method will further promote the application of SERS technology in the analysis of DNA molecules, help in the development of wearable materials for DNA detection, and offer the possibility of SERS technology to be used in disease diagnosis.

## Data Availability

The data are available upon reasonable request from the corresponding author.

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
