# Peer review of "Label-Free Detection of DNA via Surface-Enhanced Raman Spectroscopy Using Au@Ag Nanoparticles"

_nanomaterials, 2022, doi:10.3390/nano12183119_

Round 1

Reviewer 1 Report

In the manuscript “Label-Free Detection of DNA via Surface-Enhanced Raman Spectroscopy Using Au@Ag Nanoparticles” the authors show how it is possible to use Au@AgNPs core-shell structure to enhance the SERS signal in a label-free DNA detection.

In the manuscript the idea is well developed, specifically for using Au@AgNPs in SERS analysis by preserving the DNA structure. However, there are a few points that need to be addressed.

Therefore, I recommend it for publication after minor revisions are noted.
Below are the comments to the authors.

1. At line 64 of page 2 authors say “Therefore, various morphologies of nanoparticles have been designed as substrates for SERS enhancement, such as nanospheres, nanostars, nanosheets, nanoflowers, etc [38-42]” For completeness they can also mention the article:

Petti, Lucia, et al. "A plasmonic nanostructure fabricated by electron beam lithography as a sensitive and highly homogeneous SERS substrate for bio-sensing applications." Vibrational Spectroscopy 82 (2016): 22-30.

Palermo, Giovanna, et al. "Plasmonic metasurfaces based on pyramidal nanoholes for highefficiency SERS biosensing." ACS applied materials & interfaces 13.36 (2021): 43715- 43725.

2. At line 124 of page 4 add a reference on the seed growth method used.

3. At line 149 of page 4 authors say that they “…prepared dsDNA … and characterized it by circular dichroism”. No information regarding the instrument used for the CD measurements is reported. Please do the same with the DLS, TEM, and HAADF-STEM.

4. At line 180 of page 6 authors say: “It is probably due to the bimetallic synergistic effect of the Au@Ag core-shell structure that greatly enhances the SERS signal.” I think that is the crucial point of the observation. The authors can try to give a better explanation of this aspect.

5. At line 183 of page 6 authors say: “In overview, we innovatively invent a method for labelfree detection of dsDNA”. I suggest to the authors to change the verb “invent” with “observe”.

6. Figure 4 no error bar is reported on the histogram of figure b and c.

Reviewer 2 Report

In this paper the authors propose a SERS platform for DNA detection in which the Au@AgNPs core-shell structure was selected as the enhancement substrate.

The idea of using AuAg structures for SERS is not new and has been reported in several papers. Here the use of these NPs (as core-shell) for detecting DNA is technically ok

the manuscript needs significant improvements before to be accepted for publication. In particular, the english level is rather low and some sections are hard to be read and understood (for example, line 34 "but also cannot effectively restore the true information of DNA") I don't see the meaning "restore"??

From the technical point of view:

the authors report several DNA spectra, but almost completely missed to discuss the features observed in the spectra. for example in figure 4 , there is a clear difference between the spectra and the data obtained with 2.0h and no discussion is reported. also in figure 6 there are significant differences among the spectra and it can be an important improvement for the manuscript to discuss them in details

The authors must also improve the discussion including several important examples of SERS-DNA reported in literature (that they completely missed)

J. Raman Spectrosc. 1986, 17, 289−298

Angew. Chem., Int. Ed. 2015, 54, 1144−1148

Chem. Commun. 2011, 47, 10966.

Nano Lett. 2017, 17, 5071−5077

J. Phys. Chem. C 2020, 124, 41, 22663–22670

Chem. - Eur. J. 2012, 18, 5394−5400

Round 2

Reviewer 2 Report

I think that the manuscript is improved, anyway I cannot see why the authors missed to include more references with respect to the literature. The topic has been extensively explored and it is important to better introduce it. (for example some of the recommend references have been excluded for no reason)
